



# Multidecadal persistence of organic matter in soils: investigations at the submicrometer scale

Suzanne Lutfalla[1,2], Pierre Barré[2]*, Sylvain Bernard[3], Corentin Le Guillou[4], Julien Alléon[3,5], Claire Chenu[2]

[1]ECOSYS, INRA, AgroParisTech, Université Paris-Saclay, UMR 1402, Thiverval Grignon, France
[2]Laboratoire de Géologie de l'ENS - PSL Research University – CNRS UMR 8538, Paris, France
[3]Muséum National d'Histoire Naturelle, Sorbonne Université, CNRS UMR 7590, IRD, Institut de Minéralogie, de Physique des Matériaux et de Cosmochimie, Paris, France
[4]Unité Matériaux et Transformations, UMET, CNRS UMR 8207, Université de Lille, France
[5]Now at MIT - EAPS Department, Summons Lab, Cambridge MA, USA

*Correspondence to*: Pierre Barré (barre@biotite.ens.fr)

**Abstract.** The mineral matrix, particularly clay-sized minerals, protects soil organic matter (SOM) from decomposition by microorganisms. Here we report the characterization of SOM and associated minerals over decades of biodegradation, in a French long-term bare fallow (LTBF) experiment started in 1928. The amounts of carbon (C) and nitrogen (N) declined with

time for six fractions (sand, coarse silt, fine silt, coarse clays, intermediate clays and fine clays). The C:N ratios of SOM associated to silt fractions remained constant whereas they significantly decreased in clays, reaching very low values in intermediate and fine clays (C:N < 5) after 8 decades of LTBF. X-ray absorption spectroscopy revealed that (i) bulk-scale SOM chemical speciation remained almost constant, (ii) submicrometric particulate OM was present in coarse clays, even after 79 years of LTBF, (iii) illite particles became progressively SOM-free with time whereas mixed layered illite/smectite

and smectites were always associated to OM throughout the bare fallow. Altogether, these results suggest that clay-sized minerals preferentially protect N-rich SOM and that smectites and mixed layered illite/smectite protect SOM more efficiently than illites.

## 1 Introduction

Soils represent an important carbon reservoir on the global scale: they store three times more carbon than the atmosphere

(Batjes, 1996) and are currently considered as one of the solutions for climate change mitigation and adaptation and food security as highlighted by the "4 per 1000" initiative (Soussana et al., 2017). Soil organic matter encompasses compounds with residence times ranging from days to millennia (Trumbore, 2000). The mechanisms controlling SOM turnover are actively debated (Dungait et al., 2012; Lehmann and Kleber, 2015; Schmidt et al., 2011).

Except to some extent for pyrogenic C (Lutfalla et al., 2017), the current thinking considers organomineral interactions (by

adsorption or coprecipitation) as a dominant factor, rather than intrinsic chemical recalcitrance, for the long-term persistence in soil of otherwise labile organic compounds (Baldock and Skjemstad, 2000; Balesdent et al., 2000; Lehmann and Kleber,





2015; Lützow and Kögel-Knabner, 2010; Schmidt et al., 2011; Sollins et al., 2006). Yet, the exact mineralogical nature of soils remains barely documented, especially at the submicrometer scale, and in particular for soils whose clay fraction is dominated by phyllosilicates (Barré et al., 2014).

Soil phyllosilicates are very diverse (Wilson, 1999). Among phyllosilicates, smectites are considered to have higher protective
capabilities than illite and kaolinite, because of their higher specific surface area and cation exchange capacities (Bruun et al., 2010). Nonetheless, *in-situ* experimental demonstrations are still lacking (Barré et al., 2014). To date, the influence of phyllosilicate mineralogy on the chemical composition of persistent SOM has only been documented in model systems (Mikutta et al., 2007).

Here, we studied samples from long-term bare fallow (LTBF) experiments. These vegetation-free experimental plots offer the
10 unique opportunity to naturally concentrate persistent SOM (Barré et al., 2010; Rühlmann, 1999): with time, biodegradation occurs in the LTBF plots and carbon content gradually decreases as there is no input of fresh organic carbon.

We used quadruplicate LTBF soil samples collected in 1929, 1939, 1951, 1981 and 2008 and fractionated in six particle-size classes (sand [> 50 µm], coarse silt [20 – 50 µm], fine silt [2 – 20 µm], coarse clays (CC) [0.2 – 2 µm], intermediate clays (IC) [0.05 – 0.2 µm] and fine clays (FC) [0 – 0.05 µm]). The three clay subfractions contained diverse phyllosilicates (Fernández-
15 Ugalde et al., 2016; Hubert et al., 2012) which were characterized by X-ray diffraction. Synchrotron-based near-edge X-ray absorption fine structure (NEXAFS) spectroscopy at the C K edge is a pertinent tool to study SOM because it provides information on the carbon speciation regardless of the mineral matrix in which carbon particles are embedded. NEXAFS data were collected on clay fractions at the "bulk" scale ($\sim$ mm$^3$) using fluorescence-based X-ray spectroscopy and at the submicrometer scale using scanning transmission X-ray microscopy (STXM). Combined, these previous observations allowed
demonstrating the heterogeneous nature of SOM (Gillespie et al., 2015; Keiluweit et al., 2012; Kinyangi et al., 2006; Lehmann et al., 2005; Wan et al., 2007) and the complex relationship between SOM spatial distribution and soil mineral composition (Solomon et al., 2012; Wan et al., 2007).

The present contribution addresses the following fundamental questions: (1) what is the amount of pluri-decadal persistent SOM in the different fractions? (2) What is the chemical nature of pluri-decadal persistent SOM in the different fractions? And
25 (3) what is the long-term protective capabilities of the different phyllosilicates?

## 2 2. Materials and methods

### 2.1 Soil description and sampling

We used archived samples of the "*42 Parcelles*" LTBF experiment to study the evolution of SOM towards more persistent forms of SOM over 79 years of biodegradation. The "*42 Parcelles*" LTBF is an INRA (Institut National de la Recherche
Agronomique) experiment which started in Versailles (France) in 1928 (Burgevin and Hénin, 1939). Since 1929, the soil has not been cropped and has been weeded (by hand or with herbicides) and plowed (to the depth of 25 cm) twice a year. It is a non-carbonated brown soil (silty loam Luvisol: 16% clay, 57% silt, 27% sand) and the control plots have an average pH value



of 6.3 in 1929 and of 5.2 in 2008 (Grasset et al., 2009; Paradelo et al., 2013; Pernes-Debuyser and Tessier, 2002). We used archived samples (air dried and stored in the dark) from 4 plots (13, 21, 22 and 32) sampled at 5 different dates: 1929 (year 0 of the LTBF), 1939 (year 10), 1951 (year 22), 1981 (year 52) and 2008 (year 79). Samples come from the first 25 cm of soils, except the last ones (2008) that correspond to the first 20 cm.

## 2.2 Particle size fractionation

We subjected soil samples to physical dispersion and particle-size fractionation following a published protocol (Balesdent et al., 1998, Fernández-Ugalde et al., 2016) to separate the following fractions: sand fraction (> 50 µm), silt fractions (20 – 50 µm, 2 – 20 µm) and clay fractions (0.2 – 2 µm, 0.05 – 0.2 µm, and < 0.05 µm).

Approximately 50 g of soil were shaken overnight with 20 glass beads in 180 ml of deionized water to break aggregates bigger than 50 µm. The obtained suspension was then passed through a 50-µm sieve, thereby preventing the disruption of particulate organic matter during the subsequent ultrasonic dispersion (Balesdent et al., 1991), and sonicated in an ice bath for a total input of 320 J.ml$^{-1}$ using a digital sonifier (Sonics model 500W operating at 20 kHz – max. output = 120 W - probe with a flat tip of 2.5 cm diameter). The dispersion conditions allowed recovering a clay-size fraction (< 2 µm) equivalent in proportion to that achieved during standard particle-size fractionation (Balesdent et al., 1991). At that stage, the solution was centrifuged at 79g for 12 minutes to first isolate the clay fraction (0 – 2 µm) then at 5g for 1 minute to isolate the silt fractions (2 – 20 and 20 – 50 µm). Before weighing, sand and coarse silt fractions were oven-dried at 60 °C while the fine silt and clay fractions were freeze-dried. Total mass recovery was 99.1% (SD=0.44%).

Clay subsamples (2 g) were suspended in water and sonicated at 320 J.ml$^{-1}$. The sonicated suspension was then sequentially centrifuged at 23056g for 61 minutes to isolate the fraction < 0.05 µm and at 5764g for 15 minutes 22 seconds to separate the smaller fractions (0.05 – 0.2 and 0.2 – 2 µm). All clay subfractions were then freeze-dried. Mass recovery for the clay fractionation was 98.3% (SD=1.70%).

## 2.3 Clay mineralogy analysis

The mineral compositions of the total clay fraction and of the clay subfractions (CC, IC and FC) were determined by X-ray diffraction (XRD) analyses. Air-dried, oriented deposits prepared using the filter transfer method (Moore and Reynolds, 1997) were analyzed with a Cu Kα radiation RIGAKU UltraX18HF X-ray diffractometer (RIGAKU, Tokyo, Japan). The XRD patterns were accumulated at 0.05º step intervals with a counting time of 3 s in the range 3-35º (2θ). One set of replicates was then exposed to ethylene glycol vapor over 16 hours at 60ºC before being re-measured under the same conditions as the air-dried samples.




### 2.4 Elemental analysis

The total organic carbon (C) and nitrogen (N) contents of all particle-size fractions were measured by dry combustion in a CHN autoanalyser (Carlo Erba NA 1500). The total C content is equivalent to the total organic carbon content as the investigated soil samples do not contain carbonates.

### 2.5 Synchrotron-based NEXAFS spectroscopy

In the present study, synchrotron-based C-NEXAFS data were collected using beamlines located at the Canadian Light Source (CLS – Canada) whose storage ring is operated at 2.9 GeV and between 250 and 150 mA current.

### 2.5.1 C-NEXAFS spectroscopy at the "bulk" scale

The 'bulk' carbon speciation of clay subfractions was investigated by NEXAFS spectroscopy using the CLS beamline 11-ID-1 Spherical Grating Monochromator (SGM) (Regier et al., 2007). See Supplementary Material for details about the method. Each spectrum reported in the present study corresponds to the average of about 50 measurements. Of note, only the first 250 nanometers of the sample surface are probed using the SGM setup. Spectra were averaged, background subtracted and normalized using the Igor Pro software.

### 2.5.2 STXM-based NEXAFS spectroscopy

STXM-based NEXAFS data were collected using the CLS beamline 10ID-1 (SM beamline(Kaznatcheev et al., 2007) that works in the soft X-ray energy range (130–2500 eV) using an elliptically polarized undulator. See Supplementary Material for details about the method. All clay subfractions were analysed for each sampling time, i.e. 15 samples were analysed. We first analyzed each sample at the millimetric scale to identify regions of a few square micrometers on which to collect full STXM-based NEXAFS data over the 250-450 eV energy range covering the C K edge (280-295 eV), the K $L_{2,3}$ edges (295-305 eV), the Ca $L_{2,3}$ edges (345-355 eV) and the N K edge (395-405 eV). For each sample, we selected one or two regions which contained carbon-rich particles and presented a diversity of absorptions allowing us to qualitatively investigate the organomineral interactions present in each fraction at each date. The C-NEXAFS spectra shown here correspond to homogeneous areas of several hundreds of square nanometers. Extensive databases of reference C-NEXAFS spectra are available in the literature (Solomon et al., 2009). The compositional maps presented in this study are derived from the analysis of the selected micrometric regions, they are the visual representation of the assignment of each pixel of the image to the category of particle it belongs to, according to its associated NEXAFS spectra.

### 2.5.3 C-NEXAFS data deconvolution procedure

To obtain a more 'quantitative' insight on the evolution of the molecular signatures of the investigated experimental samples with increasing bare fallow duration and to be able to compare the spectra, we performed i) background subtraction, ii)





normalization to the carbon amount and iii) fit using Gaussian functions placed at fixed positions (*e.g.* 284.4 eV, quinones; 285 eV, 285.4 eV, aromatic; 285.8 eV, imines; 286.2 eV, 286.6 eV, 287.1 eV, carbonyls; 287.7 eV, aliphatics; 288.2 eV, amides; 288.6 eV, carboxylic; 289.1 eV, aldehydes; 289.4 eV, hydroxyls; 289.9 eV, aliphatics; 290.3 eV, carbonates) (Bernard et al., 2012; Le Guillou et al., 2014).

**2.6 Statistical analyses**

Statistical analyses were conducted using the free software environment for statistical computing R (http://www.r-project.org). The significance of the difference between C contents or C/N ratios was assessed using pairwise t-tests. A level of significance of P = 0.05 was considered for all analyses.

**3 Results**

**3.1 Carbon and nitrogen declines in the fractions**

Initial C and N concentrations were low and very low in the sand and coarse silt fractions, respectively, but much higher in the fine silt and clay fractions (Table 1). The IC and CC subfractions displayed an initial OC content ($35.9 \pm 2.19$ mgC.g$^{-1}$ fraction and $46.6 \pm 2.95$ mgC.g$^{-1}$ fraction, respectively) 3 to 5 times lower than that of the FC ($147.8 \pm 16.9$ mgC.g$^{-1}$ fraction). Of note, dissolved organic carbon may contribute to this high value (Balesdent et al., 1998; Tiessen and Stewart, 1983). The C and N

contents of the bulk and of each fraction decreased with increasing LTBF duration and somehow stabilized after 52 years of BF (Table 1; Figure 1). The C and N declines in the coarse silt fraction were not that clear, but these fractions had very low N and C concentrations. The low C decline (53%) compared to the N decline (74%) in the sand fraction could be explained by the presence of C-rich sand-sized coal or pyrogenic carbon (Lutfalla et al., 2017). Overall, the CC, IC and FC subfractions lost 59%, 49% and 67% of their initial C content and 52%, 36% and 48% of their initial N content, respectively. The C:N ratios of

the different fractions evolved differently with bare fallow duration, while that of the bulk soil remained roughly constant. The C:N ratio of sand fractions increased (due to the increased proportion of pyrogenic carbon), while those of clay fractions decreased (the C:N ratios of IC and FC fractions reached values as low as 4.5).

**3.2 Mineralogy of the clay subfractions**

The XRD patterns of the three clay subfractions of a bare fallow sample (plot 21) collected at t = 0 showed that, in agreement

with previous results obtained on similar soils (Fernández-Ugalde et al., 2016; Hubert et al., 2009), the CC fraction contains smectite, illite, kaolinite as well as mixed-layered illite and smectite while the IC and FC fractions only contain smectite and mixed layered illite and smectite (Figure 3). The mineral composition of each investigated clay subfraction did not evolve during the 79 years of bare fallow as indicated by XRD (data not shown). The soils studied do not contain significant amounts of iron (Fe) and aluminium (Al) minerals as evidenced by Fernández-Ugalde et al., (2013) who analysed soils from an adjacent

field in the same experimental site of Versailles.





## 3.3 'Bulk' C-NEXAFS spectroscopy

The 'bulk' C-NEXAFS spectra of clay subfractions were very similar, with spectral features attributed to four main chemical moeities (Figure 4): aromatic or olefin carbons (peak between 285 and 285.5 eV), carbonyl groups (shoulder between 286.1 and 287.1 eV), aliphatic carbons (shoulder at 287.7 eV) and carboxylic groups (intense peak at 288.6 eV). The peak at 290.3

5  eV is attributed to carbonates (Bernard et al., 2015). Deconvolution of the data allowed semi-quantitative estimation of the relative concentration of the four functional groups described above (aromatics/olefinics, carbonyls, aliphatics and carboxylics) as a function of time and revealed no major evolution except for a slight but statistically significant increase in carboyxlics moeities contained in the CC subfraction and a slight but statistically significant decrease in aliphatics in the FC subfraction (Figure S3).

## 3.4 STXM-based NEXAFS spectroscopy

### 3.4.1 NEXAFS spectra of organomineral particles

Four different types of assemblages were observed in the clay subfractions (Figure 5): (1) SOM-poor K-rich minerals, which abundance increased with bare fallow duration; (2) organomineral complexes rich in C, N and K; (3) organomineral complexes rich in C, N, K and Ca; and (4) K-poor particulate OM.

### 3.4.2 Evolution with time of organomineral particles in the three clay subfractions

Over the course of the bare fallow, CC subfractions displayed particles ranging from isolated OM (particulate OM) to mineral-rich, OM-bearing particles (Figures 5 & 6). After 22 years of bare fallow, mineral particles exempt of organic matter appeared in the CC subfractions. On the other hand, mineral particles exempt of organic matter and OM rich particles were virtually not detected in the IC and FC subfractions. These subfractions exhibited a homogeneous signal similar to the spectra of OM+K+Ca

organomineral particles, *i.e.* assemblages involving SOM and smectite or mixed layered clay particles (Figures 5 & 6).



## 4 Discussion

### 4.1 Persistent SOM in clay fractions is N-rich

After 79 years of bare fallow, SOM was remaining in all the soil fractions. Apart from the coarse silt fraction which contained very little amounts of SOM, the amount of OC and N remaining after 79 years of bare fallow were, as expected, higher in the

clay fractions (Table 1). Of note, the relatively high amount of C remaining in sand fractions could be explained by the presence of pyrogenic carbon in these fractions (Table 1, Figure 2).

The percentage of C and N remaining is increasing with decreasing particle-size for fine silt, CC and IC and then decreasing for FC suggesting a higher content of labile SOM in the FC fraction. This is likely due to the fractionation procedure which can favor the accumulation of labile dissolved OM in the FC fraction (Laird et al., 2001).

Although it was not possible to determine the nitrogen speciation, our results show that persistent SOM associated to clays is highly enriched in N. Indeed, particulate organic matter with high C:N ratios are present in coarse fractions, while smaller particle-sizes typically have lower C:N ratios (Balesdent et al., 1987; Christensen, 1992; Fernández-Ugalde et al., 2016). Interestingly, while the C:N ratios of SOM associated to silt fractions did not evolve, the C:N ratios of SOM associated to clay fractions significantly decreased with bare fallow duration (down to values as low as 4.5 for the IC and FC fractions, which

has to our best knowledge never been reported before). Accordingly, it appears that compounds with nitrogen moieties have a strong affinity for mineral surfaces (Kleber et al., 2007).

NEXAFS showed no major shift in the chemistry of SOM after several decades of biodegradation under bare fallow conditions besides a slight increase in carboxylics moieties in the CC subfraction and a slight decrease in aliphatics in the FC subfraction, supporting that persistent forms of carbon are slightly more oxidized than the initial forms of carbon (von Lützow and Kögel-

Knabner, 2010). The present study also confirmed that persistent SOM is mainly composed of microbial material: on average, all spectra displayed typical patterns of soil organic matter strongly enriched in microbial material (Keiluweit et al., 2012; Kleber et al., 2011).

### 4.2 Particulate organic matter persists in clays after decades of biodegradation

Particulate OM could still be observed in the CC subfraction even after 79 years of bare fallow. The NEXAFS spectra of these

particles highlighted their polyphenolic nature (Keiluweit et al., 2010), suggesting that these were pieces of lignin-rich plant debris, possibly physically protected in submicron aggregates as shown in similar temperate luvisols (Chenu and Plante, 2006). The present results thus show that pluri-decadal persistent SOM is made of N-rich oxidized SOM adsorbed to mineral surfaces and, to a lesser extent, of particulate OM, in agreement with the "soil continuum model (SCM)" recently proposed (Lehmann and Kleber, 2015).

A simple mixing model allows roughly estimating the amount of C associated to minerals in the CC subfractions. Assuming a C:N ratio of 10 for pure particulate OM (corresponding to C:N observed in silt fractions and referred to as $C:N_{particulate\ OM}$) and a C:N ratio of 4.5 for OM bound to minerals (referred to as $C:N_{OM-minerals}$,) *i.e.* without particulate OM (corresponding to C:N





ratio observed in IC and FC subfractions after 79 years of bare fallow), and knowing that the C:N ratio of CC after 79 years of bare fallow is 6.7 (referred to as C:N$_{CC, 79y}$ ), we can calculate the proportion of OM associated to minerals in CC by solving the simple equation:

$$C:N_{CC, 79y} = x * C:N_{particulate\ OM} + (1 - x) * C:N_{OM\text{-}minerals},$$

where C:N$_{CC, 79y}$ = 6.7; C:N$_{OM\text{-}minerals}$ = 4.5;  C:N$_{particulate\ OM}$ = 10 and x represents the proportion of particulate OM in the coarse clay fraction after 79 years of bare fallow.

Solving the equation leads to x = 0.4.  It can thus be estimated that the CC subfractions contain, at the end, approximately 40% of particulate OM and 60% of SOM associated to minerals. Therefore, in addition to particulate OM, the CC subfractions contain approximately 11.6 mgC.g$^{-1}$, under the form of OM bound to minerals, *i.e.* significantly less than the C content of the

IC subfractions. This might be explained by the relatively small specific surface area of relatively coarse clay minerals compared to that of finer ones.

### 4.3 Smectites appear more efficient at protecting SOM than illites

After 79 years of bare fallow, the FC and IC subfractions, mostly composed of swelling clays, had a higher N content compared

to the CC subfraction that contained both illites and swelling clays (Figure 3). The CC and IC subfractions had a similar apparent C content after 79 years of bare fallow, lower than that of the FC subfraction (Table 1). However, as seen above, TOC contents might be misleading in this particular case as CC contains significant amounts of POM (estimated to be around 40% of TOC in this fraction). Therefore, if we focus on C associated to clay minerals by organomineral interactions (either through adsorption or coprecipitation), we find that at all times, there is more C associated to IC and FC than to CC. These

results may be due to the predominance of swelling clays in IC and FC or to the fact that finer clay minerals have a higher specific surface area and can therefore interact with more SOM. These results also suggest that swelling clays may better protect N-rich SOM in particular.

Spatially resolved observations at the submicron scale with STXM-NEXAFS clearly showed that mineralogy influences SOM stabilization. Indeed, several illite particles (identified by the presence of K and the absence of Ca) were devoid of OM (Figures

5 & 6) and the relative abundance of OM-depleted illites increased over time. On the other hand, smectite layers (identified by the presence of Ca, as interstratified illite/smectite in the presence of K or pure smectite in the absence of K) were always found in association with OM over the chronosequence. This demonstrates that among phyllosilicates, smectites might have higher SOM protective capabilities than illites. Our results do not allow us to conclude on the protective capability of kaolinite, and additional experiments are thus needed to confirm the present findings and investigate clays with other mineralogies.

The suggested higher protective capacity of smectites to protect OM  compared to illites could be due to the presence of calcium which might facilitate the formation of persistent bounds between clay surfaces and OM. Organomineral assemblages rely on different physico-chemical interactions, depending on the chemical nature of the OM and of the mineral phase, they

include covalent bonding, ligand exchange or weaker interactions such as Van der Walls for instance (Barré et al., 2014; Kögel-Knabner and Amelung, 2014). Similarly to previous studies (Chen et al., 2014), we observed a co-location of C and Ca (contained in smectites). Although the exact mechanism responsible for this co-location is not clearly identified, one hypothesis is that Ca could facilitate the binding of negatively charged or polarized organic compounds to negatively charged mineral

surfaces via cation bridging (Lützow et al., 2006; Mikutta et al., 2007; Rowley et al., 2018). It has been shown that the cation bridging mechanism can promote organomineral interaction and reduce the bioavailability of adsorbed organic molecules for negatively charged 2:1 clay minerals (vermiculite) (Mikutta et al., 2007). In contrast to smectites, the negative charges of illite surfaces are mostly compensated by $K^+$ ions. $K^+$ ions are monovalent and have a lower charge/radius ratio and, as a result, are much less efficient to act as a bridge between organic compounds and mineral surfaces. Additionally, smectites have a higher

specific surface area and could adsorb more OM. These may explain the lower capability of illite particles to protect SOM as evidenced here.

## Acknowledgments

The INSU EC2CO program is acknowledged for financial support. NEXAFS data were acquired at the beamline11ID-1 at the CLS, which is supported by the NSERC, the CIHR, the NRC and the University of Saskatchewan. Special thanks go to Jian

Wang and Jay Dynes for their expert support of the STXM at the CLS and to Tom Regier and Adam Gillespie for their expert support on the SGM-beamline at CLS.

## Supplementary material

Materials and methods on C-NEXAFS spectroscopy at the "bulk" scale, STXM-based NEXAFS spectroscopy, C-NEXAFS data deconvolution procedure. Additional figures of Normalized NEXAFS spectra of coarse and intermediate clay subfractions

at all sampling dates (Figures S1 and S2) and corresponding deconvolution (Figures S3 and S4).

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




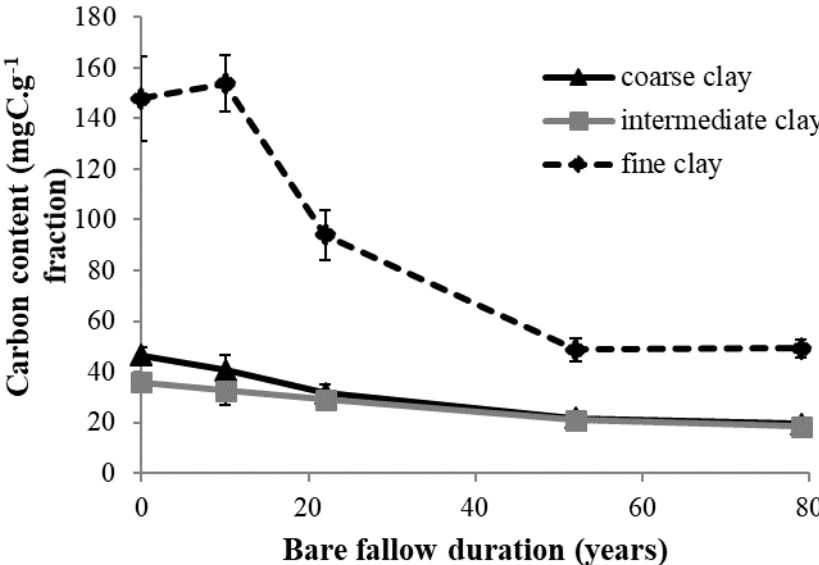

**Figure 1: Evolution of carbon content (mgC.g⁻¹ fraction) with time in the three different clay subfractions (error bars represent the standard deviation observed over the 4 field replicates).**




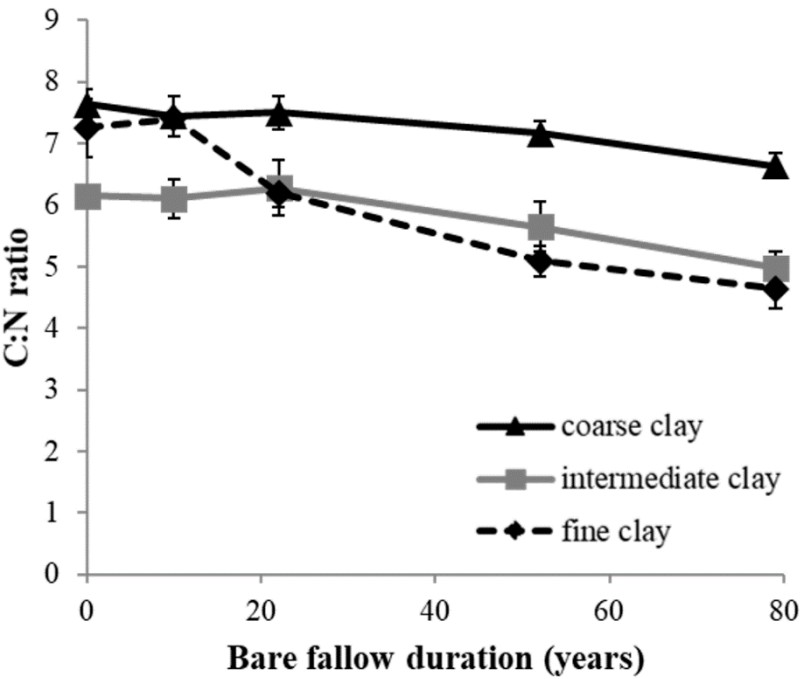

**Figure 2: Evolution of carbon to nitrogen ratio (C:N) with time in the three different clay subfractions (error bars represent the standard deviation observed over the 4 field replicates).**



| Fraction | Bulk soil | Sand | Coarse silt | Fine silt | Total clays | Coarse clays | Intermediate clays | Fine clays |
|---|---|---|---|---|---|---|---|---|
| Average % weight of each fraction | 100 | 25.2 (1.3) | 51.0 (2.7) | 8.1 (1.3) | 14.7 (1.2) | 9.8 (0.3) | 3.7 (0.3) | 1.1 (0.1) |
| Initial C content ($mgC.g^{-1}$) | 18.5 (0.5) | 16.6 (1.1) | 1.7 (0.4) | 42.1 (1.4) | 53.3 (6.6) | 46.6 (2.9) | 35.9 (2.2) | 147.8 (16.9) |
| Final C content ($mgC.g^{-1}$) | 6.3 (1.0) | 7.7 (2.4) | 1.2 (0.3) | 12.5 (2.8) | 19.0 (1.5) | 19.4 (1.2) | 18.4 (2.2) | 49.2 (3.7) |
| % of C remaining final vs initial | 34.0 (5.0) | 46.6 (12.4) | 74.1 (28.4) | 29.7 (5.7) | 36.0 (5.2) | 41.9 (4.6) | 51.3 (5.0) | 33.7 (5.8) |
| Initial N content ($mgN.g^{-1}$) | 1.9 (0.03) | 1.0 (0.1) | 0.2 (0.03) | 4.3 (0.1) | 7.1 (0.7) | 6.1 (0.5) | 5.9 (0.4) | 20.4 (1.7) |
| Final N content ($mgN.g^{-1}$) | 0.6 (0.06) | 0.3 (0.1) | 0.1 (0.02) | 1.3 (0.3) | 2.84 (0.3) | 2.9 (0.1) | 3.7 (0.4) | 10.6 (1.2) |
| % of N remaining final vs initial | 33.4 (2.8) | 26.5 (6.1) | 64.1 (22.3) | 29.6 (4.8) | 40.2 (5.2) | 48.2 (4,7) | 63.3 (3.6) | 52.9 (10.1) |
| C:N ratio initial | 9.6 (0.2) | 16.7 (1.4) | 9.2 (0.8) | 9.83 (0.1) | 7.5 (0.2) | 7.6 (0.2) | 6.1 (0.1) | 7.3 (0.5) |
| C:N ratio final | 9.8 (1.0) | 29.4 (5.8) | 10.5 (2.0) | 9.8 (0.5) | 6.7 (0.1) | 6.6 (0.2) | 5.0 (0.3) | 4.6 (0.3) |

**Table 1: Carbon and nitrogen contents and corresponding losses of organic carbon and nitrogen in each of the fractions at the first and last sampling dates. Standard deviations are indicated in brackets.**



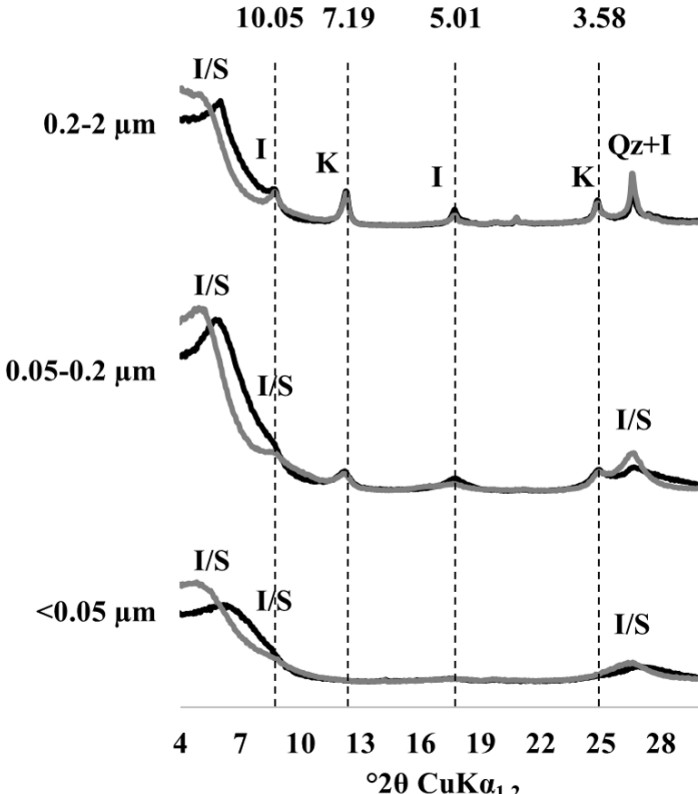

**Figure 3: X-ray diffraction patterns of particle-size fractions in one of the replicated plot of LTBF sampled at the beginning of the BF. The black line corresponds to air-dried preparations, the grey line corresponds to glycolated preparations. From top to bottom: coarse clay fraction (0.2 – 2 μm), medium clay fraction (0.05 – 0.2 μm), fine clay fraction (< 0.05 μm). Letter labels stand for the type of mineral detected: kaolinite (K), illite (I), quartz (Qz), and mixed layered illite/smectite (I/S). Numbers above the dashed lines correspond to angströms (Å).**





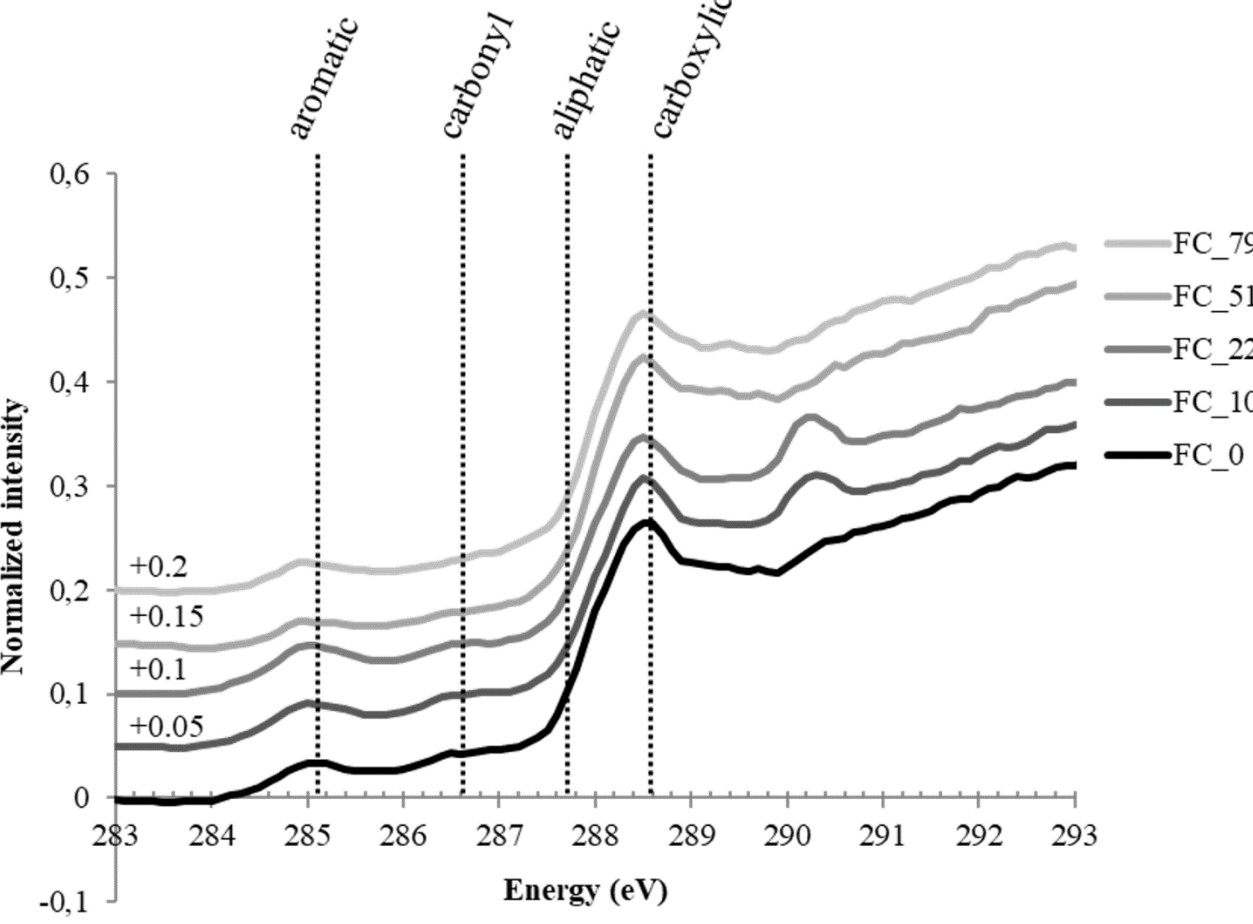

**Figure 4: Normalized C-NEXAFS spectra of the fine clay subfraction (FC) of samples collected at different times (from t = 0, bottom spectrum, darkest color, to t = 79 years of bare fallow, upper spectrum, lightest color).**





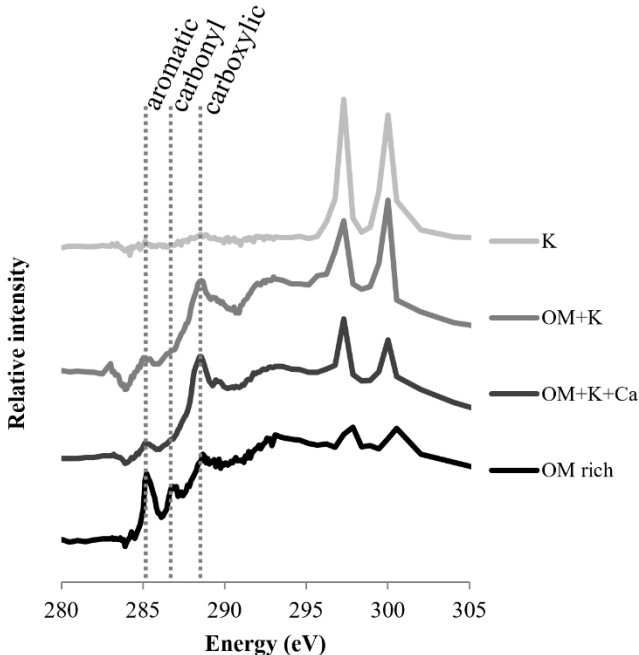

**Figure 5: Selected NEXAFS spectra of the four types of particles identified in clay subfractions by STXM. Individual spectra are shifted on the intensity axis for better discrimination. Non-labeled peaks at 297.1 and 299.7 eV correspond to Potassium (K) L edge peaks. The spectra correspond to OM-rich particles with very little mineral (darkest color), organomineral particles with K and Ca mineral phases, organomineral particles with K-minerals, and K-bearing phases (lightest color).**



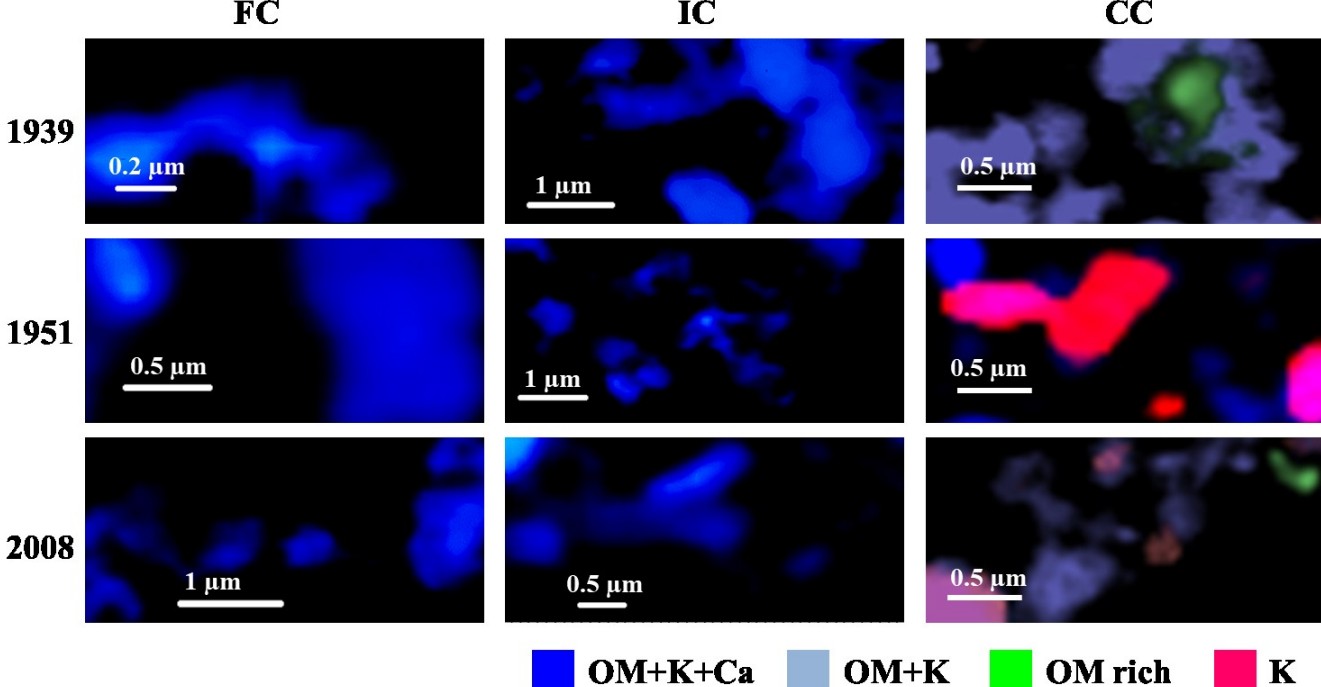

Figure 6: STXM-NEXAFS compositional maps of organomineral particle contained in the fine clay (left column), intermediate clay (middle column) and coarse clay (right column) subfractions at three different dates (from top to bottom: 1939, 1951 and the final sampling in 2008). Scale bar is represented by a white line, it varies from 0.2-1 μm and depends on the sample. Dark blue: OM+K+Ca. Light blue: OM+K. Green: OM-rich. Red/pink: K. Corresponding spectra are shown in Figure 5.