# Peer review of "Multidecadal persistence of organic matter in soils: multiscale investigations down to the submicron"

_Biogeosciences, 2018_

## Referee Comment (RC1) · Anonymous Referee #2 · 3 Oct 2018

This study addressed important issues related to multidecadal persistence of organic matter in soils at the submicrometer scale, and the results have important implications for soil carbon storage, and its mechanism of soil mineral protection. Specific comments In the discussion, it conclude that persistent SOM in clay fractions is N-rich compounds, which is supported by the C:N ratios. However, this is not solid, it should be evidenced by the N K-edge stacks or XPS information.
* * *

---

## Author Comment (AC1) · 26 Oct 2018

We thank the reviewer for his positive general comment on our draft. Regarding his/her specific comment, elemental analysis gives the C:N ratio of bulk SOM associated to the different clay fractions. We observed very substantial decreases of C:N ratios with time under bare fallow for bulk SOM associated to clays, C:N ratios reaching particularly low values (C:N below 5). We therefore concluded that persistent SOM in clay fractions is N-rich and we do not see which alternative explanation we can suggest. We agree that information based on the N K-edge stacks would have been interesting and useful. Unfortunately, as reported in the draft, for reasons we do not understand, we could not identify peaks in the N-edge region of the NEXAFS spectra. Besides, we are not aware of any study having successfully identified peaks in this region on natural SOM.

[Figure]

We fully agree that it would need further research beyond the present study.

---

## Referee Comment (RC2) · Anonymous Referee #1 · 4 Dec 2018

The topic, a potential special role of smectites in adsorbing/preserving soil OM, is still not well understood and of interest to the readers of BG. Analytical techniques with a high spatial resolution such as STXM may in fact be crucial to answer the underlying questions here. Therefore, the scientific approach is generally valid and of scientific significance. The presentation is concise and well structured. Language seems appropriate as far as I can judge as a non-native speaker. However, many details concerning data analysis and interpretation remain unclear.

The final outcome of my review is not very positive, because the main conclusion of the manuscript (smectites protect associated OM more effectively than other clay minerals) is not well supported by the given data and because the discussion of the data is rather weak:

[Figure]

P7 L23-26: A high C content in the fine clay fraction and about 100% smectite in this fraction are used to argue for a special role of smectites. This is not convincing as all investigated clay fractions are dominated by smectites. Especially the IC fraction has a much lower C-content, but only small contributions of illite and kaolinite. The smaller particle size and larger surface area of the finest clay fraction may also explain the higher C content in the FC fraction. If possible, the paper should distinguish between "dominantly smectites show an OM cover" and "smectites show a larger C-signal because they are usually much thinner than illite particles". If not possible the issue should be discussed.

Figures 5, 6, 7, i.e. the STXM stacks: The origin of the four spectra (Figure 5) is unclear. Have they been extracted from one of the stacks or are these references?

Clay mineral identification using only these C stack is unclear. All spectra of Figure 5 used to reconstruct spectra of the stacks (presumably) by linear combination fitting (LCF) show considerable contributions from K (including the "OM+K+Ca"). What is its origin, if not an illite component? Potassium does not occur in smectite and kaolinite, but in illite and mixed layer illite/smectites so that the spectrum "OM+K+Ca" can be in accordance with illite and mixed layer illite/smectite, but not with a pure smectite. However, as far as I understand the XRD data, all samples are dominated by smectites, while mixed layer phases make up for only a minor amount (if wrong, please explain!). Can K-feldspar be excluded?

Even the OM-rich fraction shows a considerable contribution of K. What is this, if not a lot of OM on illite?

Most smectites contain Ca, illites and kaolinites do not. However, it remains unclear why the spectrum "OM+K+Ca" used to represent OM-covered smectites is assumed to contain any Ca. Only at the end of the paper (P7 L29) it is mentioned that smectites were identified by the presence of Ca. HOW was that done? The Ca L-edge has its strongest peaks at 349.2 eV (L3) and 352.5 eV (L2), i.e., outside of the recorded

energy range (according to the methods sections in paper and SI).

STXM stacks are supposed to show that smectites are always associated with OM, while illites become increasingly free with the duration of bare fallow. However, only three stacks of $\sim$ 4 ïĆť 1.5 $\mu$m size are shown, two of them (2008 CC and 1951 CC) show illites (according to the LCF) with no C signal (red, but not pink areas). This is not enough to show a trend with time or mineral composition. The paper can easily accommodate a larger Figure 6/7. I suggest to show at least 10 stacks (instead of 5) in a combined figure with two columns (two size fractions) and 5 rows (sampling times). As the fine clay does not contain much illite, the IC fraction might be more interesting than the FC fraction. If there are more stacks add them to the SI.

In addition to the RGB images of components please show the residuals plus a number of fits for different ROIs. LCF will always give a result but only the residuals and the actual fits allow to judge if the result is meaningful and if the right set of components has been chosen.

I also suggest to show the calculated C concentration as separated images. This will help to visualize where OM is located with respect to the particles.

Why is the component "OM+K" not shown in Figure 6? Does not it exist? If so, please show that LCF with the three other components can reproduce the measured spectra without the "OM+K" component.

Legend of Figure 6: should not the color of "K" be red? (Please correct.) If so, only two small separate spots with that signature remain. Can you exclude that these spots are K-feldspars? At least discuss!

Figure 7: if the fine clay fraction contains only smectites (or if only smectites were imaged by STXM), how can the authors use Figure 7, to conclude that illites carry less OM than illites?

P6 L7-9: here you mention that N K-edge stacks have also been done. If so, show

them, don't withhold them! Do they fit the C/N-data? If the clay associated OM is rich in N, should not the main peak at the C K-edge be discussed as at least partly be originating from amides?

Do you have any explanation of the small peak at ~290 eV (which only appears in the OM+K+Ca component)? What is the origin of the small peak at ~283 eV and the trough at ~284 eV? What is the interpretation of the peak at ~291 eV in FC_10 and FC_22 (Figure 4)? Comparison between bulk NEXAFS data and STXM NEXAFS data is missing. How can differences be explained?

The discussion is overall rather weak. Related work is often not considered. A different interpretation (as given in the named paper by Chen) is discussed in a way that the completely different conclusions drawn from a correlation between Ca and OM by two studies does not become clear. Reasons for a potentially higher reactivity (as sorbents) of smectites relative to illites are not mentioned.

SI: I cannot judge, whether the proposed (new?) method for spectra normalization is meaningful.

---

## Author Comment (AC2) · 24 Dec 2018

Answer to the comments of Anonymous Referee 1

Notes on the font code of the text:
(1) comments from Referees (**in bold**), (2) author's response (in normal font), (3) author's changes in manuscript (*in italic*).

**The topic, a potential special role of smectites in adsorbing/preserving soil OM, is still not well understood and of interest to the readers of BG. Analytical techniques with a high spatial resolution such as STXM may in fact be crucial**

[Figure]

**to answer the underlying questions here. Therefore, the scientific approach is generally valid and of scientific significance. The presentation is concise and well structured. Language seems appropriate as far as I can judge as a non-native speaker.**

We thank the reviewer for this positive summary.

General remark on the review: it seems that some of the comments posted here by reviewer 1 are based on an earlier version of the manuscript, on which a preliminary review had been carried out at an earlier stage of the submission process. The previous comments of the anonymous referee had led us to significantly modify and hopefully improve the manuscript. Reviewer 1 might have had an access to this earlier version of the manuscript and based his or her current review on it. This might explain why most of his or her comments have already been answered and corrected in the version of the manuscript currently accessible for open online discussion and review. Indeed there are comments from reviewer 1 which do not apply to the current version of the manuscript (for instance there is no Figure 7 anymore, captions on Figure 3 have been changed, Methods section has been modified to answer the anonymous referee's comments etc). That is why we sometimes refer to the older version of the manuscript (not available for the online discussion) in our answers.

**However, many details concerning data analysis and interpretation remain unclear. The final outcome of my review is not very positive, because the main conclusion of the manuscript (smectites protect associated OM more effectively than other clay minerals) is not well supported by the given data and because the discussion of the data is rather weak:**

Firstly, we do not conclude that "smectites protect associated OM more effectively than other clay minerals" but rather that "smectites and mixed layered smectites/illites seem to protect associated OM more effectively than pure illites".

*We make this point clearer in the revised version of our manuscript.*

Secondly, our manuscript's main conclusions are not limited to the apparent differentiated OM protection provided by smectites and illites. Our study is the first to present STXM-NEXAFS data which not only allows us to describe organomineral interactions in soils but also shed light on what are the most persistent organo-mineral interactions in soils. This is due to the use of long-term bare fallow samples, some of them being 79 years old, that are fully relevant for this. Our bulk and submicrometer scale investigations led us to draw four main conclusions: (1) persistent organic matter associated to minerals is N-enriched; (2) its bulk-scale chemical speciation as seen using NEXAFS spectrocopy is not different ; (3) submicrometric particulate OM was present in coarse clays (0.2-2 $\mu$m), even after 79 years of LTBF and that (4) K-rich particles with very little OM were observed in the coarse clay fraction after several decades of LTBF whereas such OM-poor particles were not observed at the onset of the bare fallow. Given the mineralogy of this coarse fraction, we propose that such particles are particles of illite and subsequently write that these data suggest that smectites and mixed layered illite/smectite protect SOM more efficiently than illites.

We therefore consider that the reviewer is discussing one out of our four conclusions. In the following answers to the reviewer's comments we have focused on clarifying our analysis of the data presented to better explain why the data collected points towards the fact that smectites and mixed layered illite/smectite protect SOM more efficiently than pure illites.

*In addition, we suggest to change the title of the Manuscript to better reflect the multiplicity of the conclusions which involve different scales of analysis, the new proposed title is " Multidecadal persistence of organic matter in soils: multiscale investigations down to the submicron ".*

**P7 L23-26: A high C content in the fine clay fraction and about 100% smectite in this fraction are used to argue for a special role of smectites. This is not convincing as all investigated clay fractions are dominated by smectites. Especially the IC fraction has a much lower C-content, but only small contributions of illite and kaolinite. The smaller particle size and larger surface area of the finest clay**

**fraction may also explain the higher C content in the FC fraction. If possible, the paper should distinguish between "dominantly smectites show an OM cover" and "smectites show a larger C-signal because they are usually much thinner than illite particles". If not possible the issue should be discussed.**

This is right, not only the presence of smectites promotes SOM persistence, the specific surface area also plays a role. The current version of the manuscript makes this point clearer.

Additionally, STXM-NEXAFS analysis can only produce usable data when it is used on thin particles (as it is a transmission technique, thick particles are not transparent and no signal is collected), therefore the mineral particles, illites or mixed-layered smectite/illite, studied in STXM-NEXAFS are of similar thickness.

**Figures 5, 6, 7, i.e. the STXM stacks: The origin of the four spectra (Figure 5) is unclear. Have they been extracted from one of the stacks or are these references?**

The spectra shown in Figure 5 were extracted from different stacks of our data. This aspect had already been pointed out by an anonymous referee earlier in the submission process, at that time the manuscript had been modified to make this point clearer.

**Clay mineral identification using only these C stack is unclear. All spectra of Figure 5 used to reconstruct spectra of the stacks (presumably) by linear combination fitting (LCF) show considerable contributions from K (including the "OM+K+Ca"). What is its origin, if not an illite component? Potassium does not occur in smectite and kaolinite, but in illite and mixed layer illite/smectites so that the spectrum "OM+K+Ca" can be in accordance with illite and mixed layer illite/smectite, but not with a pure smectite. However, as far as I understand the XRD data, all samples are dominated by smectites, while mixed layer phases**

**make up for only a minor amount (if wrong, please explain!). Can K-feldspar be excluded?**

First of all, DRX data presented on Figure 3 show that we were not able to detect pure smectite but only interstratified illite/smectite in fine fractions (fine and intermediate clays). An earlier version of the manuscript wrongly showed the same Figure 3 with captions referring to pure smectite. This error had been pointed out by an anonymous referee at an earlier stage of the process and had been corrected in the version currently accessible online.

In addition, to answer the reviewer's comment, although there might be pure smectite on the size fractions obtained it is not predominant (as shown by DRX data) and could not be detected by STXM NEXAFS.

Finally, the samples come from a loess soil, it therefore contains very little amounts of K-feldspar and in any case not in significant amounts in the clay subfractions (it would have been detected in DRX data), it is therefore very unlikely that we are in fact looking at K-feldspar in STXM stacks.

**Even the OM-rich fraction shows a considerable contribution of K. What is this, if not a lot of OM on illite?**

As pointed out by reviewer 1, the particles showing a very strong OM signal are labelled OM rich and not OM pure, and there is still a distinguishable contribution of K in the OM rich reference spectra. This spectra is the averaged spectra of a region composed of more than ten pixels (the size of the pixel is approx. 40 nm *40 nm) and we were not able to capture a purely organic signal in our data. Therefore, minerals are still detected but in very small amounts (either illite or mixed layered illite/smectite as Ca is not shown on the graph). Nevertheless, the shape and chemical signature of these regions indicate that they resemble fresh or particulate organic matter.

*We have acknowledged the presence of minerals and added this analysis in the manuscript. Similarly, we were not able to obtain purely mineral signal, we therefore propose to change the label of particles showing a very strong K signal and a very*

*small signal for C, their label is changed to " K rich ".*

**Most smectites contain Ca, illites and kaolinites do not. However, it remains unclear why the spectrum "OM+K+Ca" used to represent OM-covered smectites is assumed to contain any Ca. Only at the end of the paper (P7 L29) it is mentioned that smectites were identified by the presence of Ca. HOW was that done? The Ca L-edge has its strongest peaks at 349.2 eV (L3) and 352.5 eV (L2), i.e., outside of the recorded energy range (according to the methods sections in paper and SI).**
Reviewer 1 might have missed the following sentences which can be found in the Methods section: "Additional image stacks have been collected at energy increments of 1 eV over the 270-450 eV energy range, allowing the rough estimation of the potassium and calcium contents. Of note, potassium and calcium are present in the interlayer and surface sites of illites and swelling clays, respectively". In an earlier version of the manuscript these sentences were located in the SI, we had moved them after a similar comment by an anonymous referee at an earlier stage of the process.

**STXM stacks are supposed to show that smectites are always associated with OM, while illites become increasingly free with the duration of bare fallow. However, only three stacks of approx. 4 * 1.5$\mu$m size are shown, two of them (2008CC and 1951CC) show illites (according to the LCF) with no C signal (red, but not pink areas). This is not enough to show a trend with time or mineral composition. The paper can easily accommodate a larger Figure 6/7. I suggest to show at least 10 stacks (instead of 5) in a combined figure with two columns (two size fractions) and 5 rows (sampling times). As the fine clay does not contain much illite, the IC fraction might be more interesting than the FC fraction. If there are more stacks add them to the SI.**
Reviewer 1 might be referring to the same " earlier " version of the manuscript. The

current version available online has been modified thanks to a similar comment made by an anonymous referee at an earlier stage of the submission process. In the current version of the manuscript more stacks have been added in what is the current version of Figure 6 (3 size fractions x 3 sampling times). Of note, the former caption of Figure 6 was misleading: 2 types of organomineral particles were observed in the data shown: OM-rich illite particles (C+K) in samples from 1939 and 2008 and interstratified Illite/smectite particles (C+K+Ca) in samples from 1951. We had made this point clearer in the current version of our manuscript. The number of stacks collected is unfortunately a limitation of the technique. Each image takes about 3h to collect and beam time is limited. That is why we do not draw definitive conclusions based on the stacks but rather propose the most parsimonious interpretation based on our data.

**In addition to the RGB images of components please show the residuals plus a number of fits for different ROIs. LCF will always give a result but only the residuals and the actual fits allow to judge if the result is meaningful and if the right set of components has been chosen.**
We performed at least 3 iterations using the "Stack Fit" procedure of the Axis software to build the RGB images shown in the Figures. Spectra that were used for the Stack Fits were those extracted from each stack after each iteration. We repeated the procedure until a residual only containing the particles too thick to be measured was obtained, i.e. until all the possible extractable information was recovered. We have illustrated an example of such a null residual in the Figure 1 of this document (stack of the CC fraction from 2008).

**I also suggest to show the calculated C concentration as separated images. This will help to visualize where OM is located with respect to the particles.**
Our data analysis procedure included a " Mapstack " analysis on the Axis software which involves the comparison by difference between two low definition images: one

taken at the Carbon K edge (energy: 288.5 eV) and one taken before the Carbon K edge (energy: 280 eV). The result is a map of C concentration, unfortunately the low definition used (the mapstacks were used to scan the samples and identify regions of interest that were then analyzed with a high resolution) prevents us from using those in the paper. However, it allowed us to verify the results of the " Stack Fit " procedure by confirming the low C concentration of the regions labeled " K rich " on RGB images presented in the paper (see below).

**Why is the component "OM+K" not shown in Figure 6? Does not it exist? If so, please show that LCF with the three other components can reproduce the measured spectra without the "OM+K" component.**
We agree with this comment, in an earlier version of the manuscript we had not differentiated K and K+Ca mineral particles for Coarse Clay subfractions. However, in the version of the manuscript currently accessible online and on which this review should be based we have changed this and now OM+K and OM+K+Ca components are clearly identified. Of note, during our data analysis, the first step of the " Stack Fit " procedure was always carried out with the four components: OM+K, OM+K+Ca, OM rich and K rich. Depending on the result of the first step and based on manual verification on the stack we performed the next steps with fewer components. It turns out that in our dataset, we did not encounter a stack presenting both K and K+Ca minerals. For Fine and Intermediate clay subfractions we detected only OM+K+Ca particles, whereas in the coarse clay subfractions we detected both OM+K and OM+K+Ca components, depending on the date and the stack.

**Legend of Figure 6: should not the color of "K" be red? (Please correct.) If so, only two small separate spots with that signature remain. Can you exclude that these spots are K-feldspars? At least discuss!**
As pointed out by reviewer 1, " pure K " regions, which would have been " red "

regions in RGB images, were very scarce. As was the case with " pure OM " regions which were not detected either (see above). This is a limitation, inherent to both the technique and the study of soils: even at submicrometric scales, and given the sample preparation procedure, we are not able to easily single out pure minerals and pure organic material.
*For this reason we now label these regions " OM rich " and " K rich ".*

Additionally, as previously explained, the soils studied contain very low to no amounts of K feldspars, it is therefore very unlikely that we are observing some in the stacks.
Furthermore, we were able to use the " Map Stacks " previously mentioned to verify that " K rich " regions were indeed poor in Carbon as can be seen in Figure 2 below.

**Figure 7: if the fine clay fraction contains only smectites (or if only smectites were imaged by STXM), how can the authors use Figure 7, to conclude that illites carry less OM than illites?**
There is no Figure 7 on that version of the manuscript, moreover fine clay fractions do not contain only smectite but mixed layered illite and smectite. Finally, our conclusion is indeed that our data suggest that illite carry less OM than smectites. To do so we use both Figure 3 (DRX) and Figure 6 which show that illite was found with very little OM whereas smectite was never found with very little OM.

**P6 L7-9: here you mention that N K-edge stacks have also been done. If so, show them, don't withhold them! Do they fit the C/N-data?**
We do not mention the N K-edge stacks in the current version of our manuscript. In fact, we have collected the data and although these stacks indicated the presence of N, it was not possible to retrieve any additional information as no peak could be observed.

**If the clay associated OM is rich in N, should not the main peak at the C K-edge be discussed as at least partly be originating from amides?**
If the samples contained a significant quantity of amide groups, peaks would be observed at 288.2 eV at the C K edge and at 401.4 eV at the N K edge. Of note, compared to an earlier version of the manuscript, the current online version displays a modified Figure 4 aimed at making the NEXAFS peak attribution clearer (zoom and additional labels).

**Do you have any explanation of the small peak at approx. 290 eV (which only appears in the OM+K+Ca component)? What is the origin of the small peak at approx. 283 eV and the trough at approx. 284 eV? What is the interpretation of the peak at approx. 291 eV in FC-10 and FC-22 (Figure 4)? Comparison between bulk NEXAFS data and STXM NEXAFS data is missing. How can differences be explained?**
The small peak at 290.3 eV in bulk NEXAFS data and STXM NEXAFS data is due to the presence of carbonates, we make this point clearer in Figures 4 and 5. The features at approx. 283-284 eV in STXM NEXAFS data are due to the presence of minerals that significantly absorb the beam, introducing noise below the edge. There are no significant differences between bulk NEXAFS data besides the additional noise in STXM-based NEXAFS data caused by the significant absorption of mineral phases.

**The discussion is overall rather weak. Related work is often not considered. A different interpretation (as given in the named paper by Chen) is discussed in a way that the completely different conclusions drawn from a correlation between Ca and OM by two studies does not become clear. Reasons for a potentially higher reactivity (as sorbents) of smectites relative to illites are not mentioned.**
We had had a similar comment from an anonymous referee at an earlier stage of the submission process and had modified the discussion to strengthen it. Of note,

the present study is in quite good agreement with the study of Chen et al. (2013) who concluded that their results "imply an important role of calcium in organo-mineral associations".

We believe that we do mention the potential differences between smectites and illite in our manuscript when we write " one hypothesis is that Ca could facilitate the binding of negatively charged or polarized organic compounds to negatively charged mineral surfaces via cation bridging (Lützow et al., 2006; Mikutta et al., 2007; Rowley et al., 2018). It has been shown that the cation bridging mechanism can promote organomineral interaction and reduce the bioavailability of adsorbed organic molecules for negatively charged 2:1 clay minerals (vermiculite) (Mikutta et al., 2007). In contrast to smectites, the negative charges of illite surfaces are mostly compensated by K+ ions. K+ ions are monovalent and have a lower charge/radius ratio and, as a result, are much less efficient to act as a bridge between organic compounds and mineral surfaces. Additionally, smectites have a higher specific surface area and could adsorb more OM. These may explain the lower capability of illite particles to protect SOM as evidenced here. "

**SI: I cannot judge, whether the proposed (new?) method for spectra normalization is meaningful.**

We applied the normalization method proposed and validated by Le Guillou et al. (2018). This normalization allows robust comparison of spectra and limits the impact of potential beam damage. Here is the full reference: Le Guillou, C., Bernard, S., De la Pena, F. and Le Brech, Y.: XANES-Based Quantification of Carbon Functional Group Concentrations, Anal. Chem., 90(14), 8379–8386, doi:10.1021/acs.analchem.8b00689, 2018.

[Figure]

[Figure]

Maximum correlation value for the
residuals.

4 reference spectra used as input
For the stackfit.

Extracted spectra corresponds to
A particle too thick and therefore
not transparent in STXM NEXAFS
(unusable)

Part of the stack used for Figure 6 in
paper.

**Fig. 1.**

[Figure]

Part of the image used in Figure 6 of the paper (1951 CC). We see that highest amounts of carbon are located in three spots (whitest regions).

The regions circled in red correspond to the Red/Pink spots on the Figure of the paper, they indeed correspond to areas with little amounts of C. Their former label read « K » we have modified it in the revised verison of the manuscript and it is now « K rich », which is more correct as we cannot say that it contains no C at all.

**Fig. 2.**

---

## Author Response (AR1)

Authors' response concerning the manuscript

**Multidecadal persistence of organic matter in soils: multiscale investigations down to the submicron**

5  Suzanne Lutfalla, Pierre Barré, Sylvain Bernard, Corentin Le Guillou, Julien Alléon, Claire Chenu

We are submitting a modified version of our manuscript following the editor's decision of « Major revisions » and the associated report « Please incorporate the suggestions of the reviewers, especially of
10  the reviewer #1 and your ms will be accepted ».
We have therefore included all the suggestions proposed in our answers to reviewers #1 and #2. As explained in our answer which can be found in the online discussion of our manuscript (https://www.biogeosciences-discuss.net/bg-2018-343/#discussion ) most of the comments made by reviewer #1 had already been incorporated in the manuscript during an earlier stage of the revision
15  process (additional images presented in the Figures which were re-organized, clarification of the materials and methods section, modification of the supplementary material, rephrasing of the results, re-organization and strengthening of the discussion, additional references …).

Here are the main changes of the present manuscript compared to the manuscript sent to the reviewers :
20  - TITLE: Modification of the title to show the diversity of tools and methods used to study the samples.
   - ABSTRACT: We have tuned down our conclusive sentence to follow the advice of reviewer #1.
   - MATERIALS AND METHODS: We have clarified the section on STXM-based NEXAFS spectroscopy and added a couple of reference to better justify our experimental setup.
25  - RESULTS:
       o We have added a couple of sentences in the section on 'Bulk' C-NEXAFS spectroscopy to answer a question of reviewer #1
       o We have lengthened a subsection in the STXM-based NEXAFS spectroscopy section to better explain our results concerning the reference spectra used in the paper, following
30          several comments made by reviewer #1
   - DISCUSSION: We have clarified our interpretation of the results in the section « Smectitic clays appear more efficient at protecting SOM than pure illites »
   - FIGURES: We have modified Figures 4, 5 and 6 following the advice of reviewer #1 (additional labels and legends to improve the readability and clarify the results).
35

Marked-up manuscript with apparent changes:

[revised manuscript text omitted]